# GDF15 Targeting for Treatment of Hyperemesis Gravidarum

**DOI:** 10.3390/medicines11070017

**Published:** 2024-08-30

**Authors:** Jamie Thygerson, Dallin Oyler, Jackson Thomas, Brandon Muse, Benjamin D. Brooks, Jessica E. Pullan

**Affiliations:** 1Department of Chemistry and Physics, Southern Utah University, Cedar City, UT 84720, USA; 2Department of Clinical Education, Rocky Vista University, Ivins, UT 84738, USA; brandon.muse@rvu.edu (B.M.); bbrooks@rvu.edu (B.D.B.)

**Keywords:** hyperemesis gravidarum, severe nausea and vomiting, GDF15, pregnancy

## Abstract

Nausea and vomiting during pregnancy (NVP), particularly its severe form, Hyperemesis gravidarum (HG), affects up to 70% of pregnancies and significantly impacts the quality of life for those with the condition as well as generates a great economic burden, with annual costs exceeding $1.7 billion in the United States. Despite the available treatments targeting neurotransmitters like serotonin and dopamine, many patients experience inadequate relief and suffer from severe side effects, including headaches and dizziness. Recent research has underscored the role of GDF15, a protein mainly produced by the placenta and linked to NVP symptoms. This protein, part of the TGF-*β* superfamily, has been implicated in appetite and weight regulation and is altered in those with HG due to specific genetic mutations. Addressing the challenges of delivering effective treatments, current innovations focus on targeting GDF15 to reduce symptoms while ensuring fetal safety. Promising therapeutic strategies include non-IgG immunotherapies, small peptide and molecule antagonists, and novel administration methods such as transdermal patches. These approaches aim to optimize dosage and reduce adverse effects. The effective development and testing of these treatments necessitate advanced animal models that closely resemble human pregnancy physiology, highlighting the need for further research and funding. This ongoing research holds significant potential to improve the clinical outcomes for HG patients and decrease the economic impact on healthcare systems, urging a dedicated response from the scientific and medical communities to advance these promising treatments.

## 1. Introduction

Nausea and vomiting are common complications that occur in 70% of pregnancies [1]. Hyperemesis gravidarum (HG) is the most severe form of these symptoms and is estimated to be prevalent in 0.3–2% of pregnancies [2]. Not only does HG impair the quality of life for pregnant mothers, but it also has significant financial ramifications for patients and medical providers. The cost of treatment for those hospitalized due to HG includes hospital admissions, intravenous hydration, nutritional support, and antiemetic medications. Recent studies indicate that the economic burden related to HG hospitalizations costs over 1.7 billion dollars annually in the United States, signaling the need for more effective treatment for HG patients [3].

Current medications that are used to treat HG include Promethazine, Clonidine, Metoclopramide, Zofran, Mirtazapine, and Corticosteroids, which regulate the neurotransmitters serotonin, dopamine, and histamine [4]. Despite these available treatments, many of those with HG continue to experience persistent and debilitating symptoms. The lack of symptom relief causes a significant physical and emotional toll on HG patients, often leading to repeated hospital admissions, the increased need for parenteral nutrition, and the inability to complete activities of daily living [5]. In addition to symptoms from HG, the potential side effects of the current medications used to treat HG, such as headaches, drowsiness, dizziness, irregular heartbeat, and confusion, further diminishes the quality of life for these patients [6,7]. The repeated hospital admissions, side effects of antiemetic drugs, and lack of symptom relief for the HG patients underscores the importance of developing effective anti-nausea medications that target the true pathway of nausea and vomiting during pregnancy (NVP). 

Effective treatment for pregnant women with NVP depends on understanding its root causes. Research indicates that increases in GDF15, often produced by the fetus rather than the mother, and maternal sensitivity are key contributors to NVP [1,8,9]. These studies also reveal that mothers carrying a mutated form of the GDF15 gene, with the fetus having a wild-type gene, face the highest risk of developing HG symptoms [1,10,11]. This highlights the critical role of both the maternal and fetal genetic factors in HG’s pathophysiology and opens avenues for developing more targeted treatments. In this discussion, we delve into the physiology of GDF15 and its implications for drug development aimed at alleviating NVP symptoms.

## 2. Physiology of GDF15

Recent research has shown a link between the upregulated levels of the protein GDF15 and the symptoms of nausea and vomiting [11,12,13]. This upregulation was determined to be influenced by mutations in the area of and surrounding the GDF15 gene locus [10]. GDF15 expression appears to be lower in mothers who experience HG prior to pregnancy; the increased GDF15 levels from the placenta appears to cause HG symptoms [10,11]. When expressed, the mature GDF15 protein contains 112 amino acid residues and forms a dimer [14]. This protein is a member of the transforming growth factor beta (TGF-*β*) superfamily, which is highly conserved among animal species and is involved in the regulation of appetite, weight, and skeletal muscle mass in humans [10]. In healthy human individuals, GDF15 is secreted in the placenta or prostate depending on sex, and trace amounts are found in other organs such as the bladder, kidney, colon, stomach, liver, gallbladder, pancreas, and endometrium [14]. Its receptor, glial-derived neurotrophic factor (GRFAL), is located in the area postrema of the brainstem and forms a complex with RET (Figure 1) [15]. The location of the GFRAL–RET complex in the brainstem does not possess a functional blood–brain barrier, allowing the circulating GDF15 and other agents to directly reach the neurons in the area postrema of the nucleus of the solitary tract [15].

## 3. Drug Development for HG

Given the complex nature of human pregnancies, a carefully balanced approach is essential in order to develop an effective medication that targets GDF15 while also ensuring fetal safety. While GDF15 deficiency can impair trophoblast invasion, which is a critical process for placental development [16], recent studies have shown that a deficiency in GDF15 could lead to the miscarriage of human embryos [16]. Conversely, higher amounts of GDF15 could lead to further symptoms of NVP, predisposing the mother and fetus towards more severe health consequences. Thus, developing a GDF15-targeted biomolecule presents many challenges that must be considered. Furthermore, targeting GDF15 in the brainstem’s GFRAL without crossing the placental barrier adds another layer of complexity to drug design [17,18]. Innovative therapeutic approaches are being explored, including the use of pharmaceuticals initially developed for obesity and cancer that target GDF15 [17,19]. Potential strategies include immunotherapies, small peptide antagonists, and small molecule antagonists, each offering unique mechanisms of action and challenges in delivery and efficacy (Figure 2).

Non-IgG immunotherapies can selectively modulate the immune system. They would present a promising approach for pregnant women with HG as this form of therapy would potentially offer an effective treatment with minimal risk to the fetus. Cancer has been a model system for this treatment strategy; however, it does not take into account the fetus. A promising IgG1 monoclonal antibody, 3P10, acts as a GFRAL antagonist, decreasing cancer cachexia [18]. This study lacks a biodistribution study within any animal models, decreasing the probability of effectiveness as an HG treatment [18]. Modifying this IgG antibody to an IgA antibody has potential as it would no longer cross the placental barrier [20].

Small peptide antagonists have also shown promise as a treatment strategy for decreasing GDF15 binding to GFRAL, ultimately reducing symptoms [15]. The peptide GRASP binds to GFRAL after GDF15 binds, preventing the attachment of RET as well as allowing for the recruitment of metal ions [15]. In rat models, GRASP demonstrated intraperitoneal injection efficacy only if delivered prior to the chemotherapeutic, cisplatin [15]. While delivery before GDF15 increases would be a feasible approach for cancer cachexia, it is less feasible for HG treatment. There could be ways to modify this peptide to increase binding affinity toward GFRAL over GDF15 in the CNS, which would increase the viability for HG use. Additional drawbacks of this strategy include overdosage, clearance, elimination, and biodistribution. 

One way to work around an injectable route of administration would be to create a small molecule pharmaceutical that could be delivered orally or as a transdermal patch. One of the largest benefits of oral delivery would be flexible dosages, allowing for a more individualized dosage. The most difficult part of an oral delivery would be bypassing the first pass effect and increasing absorption in the GI tract. A transdermal patch would also allow for more individualized dosing as the patient could remove the patch at the onset of negative symptoms. While a small molecule strategy holds promise, small molecules are more likely to cross biological barriers, including BBB and placental. As such, there are no currently published small molecule options. Targeting the hydrogen bonding within the active site of GFRAL or the exterior interaction of GFRAL to RET would be ideal. Additionally targeting GDF15 in circulation to prevent binding to GFRAL or crossing of the BBB could be another viable option. These small molecules could also be incorporated into a targeted nanoparticle, synthetic or biological, to allow for the controlled release of the therapeutic to allow for decreased placental crossing.

Overall, there are several feasible strategies to reduce the role GDF15 plays in HG/NVP (Table 1). Testing drug candidates can be tricky as the pregnancy animal models are lacking in direct physiological relationships to human pregnancy. Animal models like mice, rats, rabbits, sheep, and guinea pigs have served as model systems for pregnancy-related studies and can provide an advantageous framework for investigating the effects of GDF15 modulation [21,22,23]. However, more recent studies have shown the lack of a significant increase in GDF15 expression in mice and rats compared with macaques and humans [13,24]. With their size and extended gestation periods, sheep allow for a detailed study of fetal development and maternal interactions [21,22]. Guinea pigs, known for their placental structure similarity to humans, offer valuable insights into placental biology and fetal development [21,22]. It is likely though that many small animal species are not sufficient to draw conclusions on drug treatment options and the underlying causation of NVP. Developing these animal models, particularly for conditions like HG, is crucial for testing new treatments and understanding the condition’s pathophysiology in a controlled environment. However, it is critical to note that safety and effectiveness must ultimately be confirmed in pregnant humans. Despite advancements in animal models, direct physiological comparisons to human pregnancy remain challenging.

To ensure a comprehensive review of the potential therapeutic strategies for HG/NVP, it is worth noting that some have proposed the approach of increasing GDF15 levels prior to pregnancy to desensitize patients to the rapid rise during gestation. This approach, which could involve the use of extended-release metformin—a medication already utilized to enhance fertility in patients with PCOS due to its GDF15-elevating properties—or Lilly’s GDF15 analog, warrants further study [1].

## 4. Conclusions

The recent finding of the relationship between HG and the protein GDF15, as well as findings in GDF15 production and function, open pathways to understanding and potentially gaining more effective treatments for HG. However, targeting GDF15 in pregnant women presents unique challenges due to the complex physiology of pregnancy and safety considerations for both the mother and the developing fetus. Innovative approaches, such as immunotherapies, small peptide antagonists, and small molecule antagonists, are being explored. To test these treatment approaches, animals like sheep and guinea pigs could serve as potential pregnancy models, although further developed animal models are necessary to match human GDF15 physiology. The emerging research in the development of a biomolecule to regulate GDF15 shows promising results, as it holds the potential to effectively improve symptoms of NVP in women with HG and is opportune for further exploration and funding. With the emerging understanding of GDF15’s role in HG, this investment is crucial for advancing our knowledge and therapeutic options, ultimately improving the lives of countless women affected by this debilitating condition and reducing the economic burden on healthcare systems globally. Future studies should also explore screening methods for identifying women at risk for HG, which could enable early intervention and improve clinical outcomes.

## Figures and Tables

**Figure 1 medicines-11-00017-f001:**
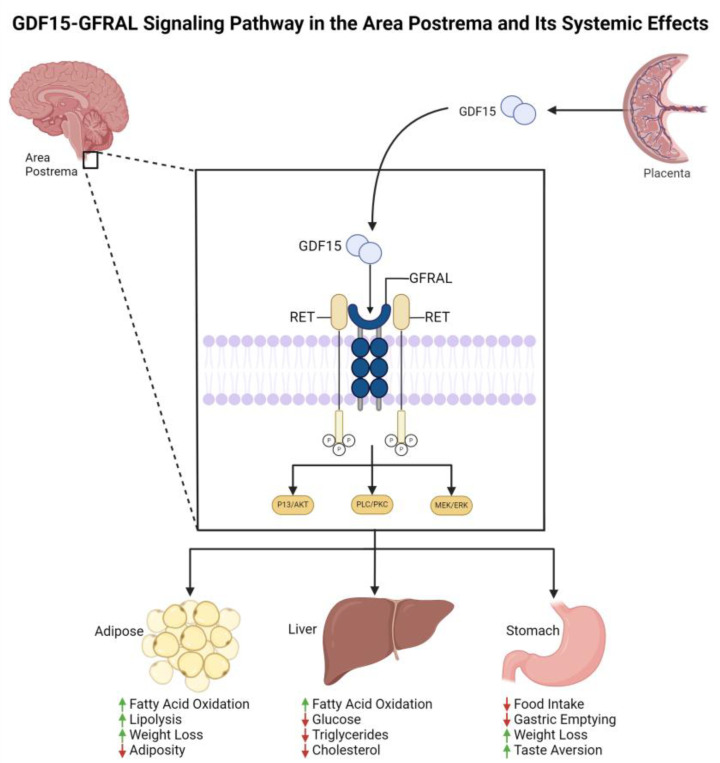
The mechanism of action of GDF15 and GFRAL in the brain. GDF15 from the placenta enters the area postrema to bind GFRAL in the area postrema, activating RET and downstream pathways (PI3K/AKT, PLC/PKC, MEK/ERK). This signaling impacts adipose tissue (increasing fatty acid oxidation and weight loss), the liver (reducing glucose and cholesterol), and the stomach (decreasing food intake and gastric emptying, inducing weight loss and taste aversion). Green arrows indicate increase, red arrows indicate decrease.

**Figure 2 medicines-11-00017-f002:**
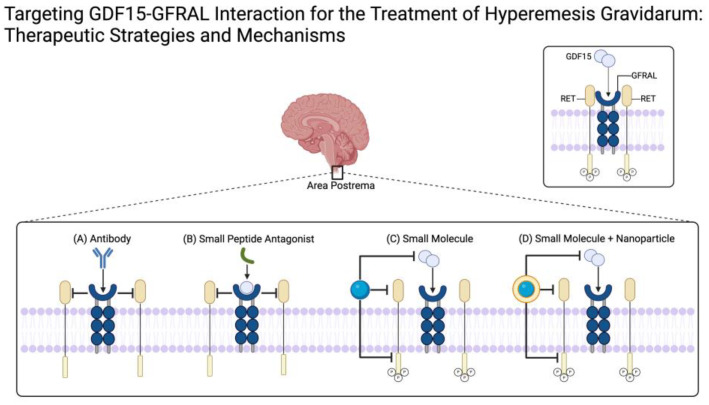
Illustrates the inhibition of GDF15–GFRAL binding and subsequent RET activation. The top section highlights the area postrema in the brain, where GDF15 binds to GFRAL, leading to RET activation (shown in detail in the top box). The bottom box details four therapeutic strategies: (**A**) an antibody that binds to GFRAL, blocking GDF15 interaction and preventing RET activation; (**B**) a small peptide antagonist that binds to GFRAL after GDF15 binds, preventing the attachment of RET; (**C**) a small molecule inhibitor that binds directly to GDF15, preventing its interaction with GFRAL, or binding to and inhibiting GFRAL and RET externally; and (**D**) a small molecule delivered via a nanoparticle to enhance targeting and reduce side effects. Each strategy aims to disrupt the GDF15–GFRAL interaction in a targeted approach.

**Table 1 medicines-11-00017-t001:** GDF15-Targeted Drugs in Development or Trials.

Drug Name	Mechanism of Action	Advantages	Disadvantages	Type of Drug
3P10 Monoclonal Antibody	GFRAL antagonist, prevents GDF15 from binding to its receptor in the brainstem.	Potentially high specificity, promising for targeted therapy with minimal side effects.	Still in experimental stages, high risk in pregnant patients (may need to change isotype).	Monoclonal Ab (IgG1)
GRASP Peptide Antagonist	Binds to GFRAL after GDF15, preventing attachment of RET and subsequent signaling.	Could provide a targeted approach with reduced systemic effects.	Limited efficacy data; challenges with dosage and biodistribution need to be addressed.	Peptide Antagonist
Unspecified Small Molecule Inhibitors	Inhibits GDF15 or GFRAL directly, blocking the signaling pathway involved in nausea and vomiting.	Flexible delivery options, including oral or transdermal administration; potential for individualized dosing.	The risk of crossing biological barriers like the placenta and blood-brain barrier is still under investigation.	Small Molecule
GDF15 Analog (e.g., Lilly’s)	Mimics GDF15 to desensitize patients to the rise in GDF15 levels during pregnancy.	Reducing the severity of symptoms by desensitization or increased affinity.	Experimental approach; potential risks to fetal development need thorough investigation.	Biologic Analog
Neutralizing GDF15 Antibody (e.g., 3D1)	Neutralizes GDF15 in circulation, preventing it from binding to GFRAL and initiating the signaling pathway.	Targeted approach, potentially reducing GDF15-induced nausea without affecting other pathways.	Still in experimental stages; safety and efficacy need to be established in human trials. Isotype is important.	Monoclonal Ab
Metformin (Extended-Release)	Increases GDF15 levels, potentially desensitizing patients to rapid rise during pregnancy.	GRAS in pregnant patients and is affordable and already used to improve fertility in PCOS patients.	Requires further research to determine efficacy for use in pregnancy.	Small Molecule

## Data Availability

No new data were created or analyzed in this study. Data sharing is not applicable to this article data.

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
