# Peer review of "GDF15 Targeting for Treatment of Hyperemesis Gravidarum"

_medicines, 2024, doi:10.3390/medicines11070017_

Round 1

Reviewer 1 Report

Comments and Suggestions for Authors

This is an important paper highlighting the recent discovery of the role GDF15 plays in HG and the great potential for novel therapies. I recommend to address the errors, edits, and suggestions below:

1.     Abstract:

a)     I would suggest replacing “reproductive organs” with “placenta”

b)    I would suggest replacing “imbalanced” with “altered” or more specifically and accurately “blood levels are altered due to specific genetic variations associated with HG.”

2.     Introduction:

a)     The statement in lines 52-54 is not quite accurate. I suggest changing to “Research indicates that maternal sensitivity to the increase in GDF15 produced by the placenta/fetus is a key contributor to the level of NVP.”

b)    The next sentence “These studies also reveal…reference 10 is the reference that identified the mutation in the mother, but not the fetus. For accuracy, I would include both reference 1 and 10.

3.     Physiology of GDF15:

a)     L60-61. Reference 11 is the study that first established a genetic link and is the wrong reference for upregulation. The paper that first showed upregulation of GDF15 in HG patients is https://pubmed.ncbi.nlm.nih.gov/31000883/

b)    L61-62. The role of the genetic variations was elucidated in the Nature paper you included in reference 1 so I am not sure why you say “is yet to be elucidated.” The variations in GDF15 associated with HG were found to be associated with lower levels of GDF15.

c)     L72-73. The paper referenced states “Importantly, these two adjacent structures do not possess a functional blood−brain barrier, allowing circulating GDF15 and other agents(including systemically delivered substances) to directly

reach the neurons located in the AP/NTS,” so I’m not sure why the authors cite it to say “a highly permeable blood-brain barrier” which is not the same as no possessing a blood-brain barrier as stated in the reference. Please fix, and also fix in the accompanying figure that shows a barrier and legend that says it crosses the barrier to get to GFRAL-this is incorrect.

4.     Drug Development:

a)     What is the data to support the statement that “GDF15 is involved in guiding the blastocyst to implant in the uterus and in spurring the development of the placenta. I’m not sure this is true, nor the next statement that deficiency could lead to miscarriage, given the recent publication of human knockouts of GDF15 are viable and fertile in addition to rodents. And women who are heterozygous for a GDF15 knockout and whose offspring also carry that knockout, presumably producing half the level of GDF15, have health viable pregnancies as shown in reference 1. If reference 14 suggests this, I would be a little more critical of this conclusion and temper down your statements. In addition, you should also discuss the KO studies that do not support these statements.

b)    I would add rabbits as a model in your list and discussion of mice, rats, etc.

c)     L. 154 I would not say “is crucial.” Regardless of any study done in animals, we cannot confirm safety and effectiveness until the studies are done in pregnant humans. Currently women are still dying from HG (it was reported to be the 4th leading cause of maternal death in Botswana in 2019), and fetus are lost and reports keep coming out of associations with adverse child outcomes including several this year, so in my opinion attempting to create an animal model of pregnancy nausea and vomiting is a waste of resources-the drug should be tested in animals to determine whether it effectively blocks the appetite and weight loss associated with GDF15-induced nausea and vomiting and also be tested using standard animal models already in use for reproductive toxicology studies. Then, if those studies show evidence of safety and effectiveness, we should move into human trials.  

This article makes no mention of the other approach suggested by the discovery published in reference 1 regarding increasing levels of GDF15 PRIOR to pregnancy to desensitize patients to the rapid rise of GDF15 during pregnancy. I think if you want to be thorough, that approach should at least be mentioned, if not discussed, with the possibility of using the relatively affordable medication, extended release metformin (that increases GDF15 and is already used in this timeframe to improve fertility in patients with PCOS), and/or Lilly’s GDF15 analog for this strategy.

Author Response

We sincerely thank the reviewers for their thoughtful and constructive feedback on our manuscript. Their insights have greatly enhanced the quality and clarity of our work, and we have carefully addressed each of their comments in our revisions. We appreciate the opportunity to improve our manuscript and are confident that these changes have strengthened the overall contribution of our study.

Comment 1:     Abstract:

  1. a)     I would suggest replacing “reproductive organs” with “placenta”

Response: We have made the suggested change on line 14.

  1. b)    I would suggest replacing “imbalanced” with “altered” or, more specifically and accurately, “blood levels are altered due to specific genetic variations associated with HG.”

Response: We have made the suggested change to altered on line 15.

Comment 2:     Introduction:

  1. a)     The statement in lines 52-54 is not quite accurate. I suggest changing to “Research indicates that maternal sensitivity to the increase in GDF15 produced by the placenta/fetus is a key contributor to the level of NVP.”

Response: We have changed the sentence to read: “Research indicates that increases in GDF15, often produced by the fetus rather than the mother, and maternal sensitivity are key contributors to NVP”

  1. b)    The next sentence, “These studies also reveal…reference 10 is the reference that identified the mutation in the mother, but not the fetus. For accuracy, I would include both references 1 and 10.

Response: This reference has been added.

Comment 3:     Physiology of GDF15:

  1. a)     L60-61. Reference 11 is the study that first established a genetic link and is the wrong reference for upregulation. The paper that first showed upregulation of GDF15 in HG patients is https://pubmed.ncbi.nlm.nih.gov/31000883/

Response: This additional publication had been previously cited in the article and has been cited here additionally.

  1. b)    L61-62. The role of the genetic variations was elucidated in the Nature paper you included in reference 1 so I am not sure why you say “is yet to be elucidated.” The variations in GDF15 associated with HG were found to be associated with lower levels of GDF15.

Response: We have reworded that statement to say: “This upregulation was determined to be influenced by mutations in the area in and surrounding the GDF15 gene locus.[10] GDF15 expression appears to be lower in mothers who experience HG prior to pregnancy; the increased GDF15 levels from the placenta appear to cause HG symptoms.[10,11]”

  1. c)     L72-73. The paper referenced states, “Importantly, these two adjacent structures do not possess a functional blood−brain barrier, allowing circulating GDF15 and other agents(including systemically delivered substances) to directly reach the neurons located in the AP/NTS,” so I’m not sure why the authors cite it to say “a highly permeable blood-brain barrier” which is not the same as no possessing a blood-brain barrier as stated in the reference. Please fix, and also fix in the accompanying figure that shows a barrier and legend that says it crosses the barrier to get to GFRAL-this is incorrect.

Response: Thank you to the reviewer for catching our error. This has been updated to discuss the area postrema.

Comment 4:     Drug Development:

  1. a)     What is the data to support the statement that “GDF15 is involved in guiding the blastocyst to implant in the uterus and in spurring the development of the placenta? I’m not sure this is true, nor the next statement that deficiency could lead to miscarriage, given the recent publication of human knockouts of GDF15 are viable and fertile in addition to rodents. Women who are heterozygous for a GDF15 knockout and whose offspring also carry that knockout, presumably producing half the level of GDF15, have health-viable pregnancies, as shown in reference 1. If reference 14 suggests this, I would be a little more critical of this conclusion and temper down your statements. In addition, you should also discuss the KO studies that do not support these statements.

Response: This has been shown in this citation: “Zeng, Y.-T.; Liu, W.-F.; Zheng, P.-S.; Li, S. GDF15 Deficiency Hinders Human Trophoblast Invasion to Mediate Pregnancy Loss through Downregulating Smad1/5 Phosphorylation. iScience 2023, 26, doi:10.1016/j.isci.2023.107902.” We have slightly reworded that section.

  1. b)    I would add rabbits as a model in your list and discussion of mice, rats, etc.

Response: We have added rabbits to the list of animal models. We did not find any studies that use rabbits as a model for GDF15 and pregnancy. We have included one citation from 2012 about rabbits being a feasible model for pregnancy but no other current literature was found.

  1. c)     L. 154 I would not say “is crucial.” Regardless of any study done in animals, we cannot confirm safety and effectiveness until the studies are done in pregnant humans. Currently, women are still dying from HG (it was reported to be the 4thleading cause of maternal death in Botswana in 2019), and fetuses are lost, reports keep coming out of associations with adverse child outcomes, including several this year, so in my opinion, attempting to create an animal model of pregnancy nausea and vomiting is a waste of resources-the drug should be tested in animals to determine whether it effectively blocks the appetite and weight loss associated with GDF15-induced nausea and vomiting and also be tested using standard animal models already in use for reproductive toxicology studies. Then, if those studies show evidence of safety and effectiveness, we should move into human trials.  

Response: We understand the reviewer’s perspective and agree with the urgency of the matter. However, we consider it unlikely that drugs would be approved, especially in pregnant persons, without preclinical studies, including animal work. Animal work is a critical piece prior to moving into human studies. Again, we agree with the urgency of the matter. We have reworded that section to ensure we highlight the importance of examining human studies for all drug development. However, we do not find it to be a waste of resources, but it provides a necessary piece in studying efficacy, toxicology, etc, prior to moving into pregnant human models not only from an ethical perspective but also from a regulatory perspective.

Comment 5:  This article makes no mention of the other approach suggested by the discovery published in reference 1 regarding increasing levels of GDF15 PRIOR to pregnancy to desensitize patients to the rapid rise of GDF15 during pregnancy. I think if you want to be thorough, that approach should at least be mentioned, if not discussed, with the possibility of using the relatively affordable medication, extended-release metformin (that increases GDF15 and is already used in this timeframe to improve fertility in patients with PCOS), and/or Lilly’s GDF15 analog for this strategy.

Response:  We agree. We have added a paragraph addressing this point (L 160-164).

Reviewer 2 Report

Comments and Suggestions for Authors

Your study is about HG, a worldwide associated pregnancy issue. The direction of your study targeting a possible association with GDF15 protein is interesting for readers and also for practicing clinicians. 
Even if your personal contribution is unclear a comprehensive awareness of this subject could raise interest. 
To improve your manuscript you should focus in introduction more on the current knowledge of HG and any other proteins and what treatment is effective based on EBM. 
Regarding paragraph 3 - drug development - a table could be more useful to synthetise already known information regarding treatment options

in conclusion section you could also address study directions relating to screening women at risk for HG.

Author Response

We sincerely thank the reviewers for their thoughtful and constructive feedback on our manuscript. Their insights have greatly enhanced the quality and clarity of our work, and we have carefully addressed each of their comments in our revisions. We appreciate the opportunity to improve our manuscript and are confident that these changes have strengthened the overall contribution of our study.

possible association with GDF15 protein is interesting for readers and also for practicing clinicians. 
Even if your personal contribution is unclear a comprehensive awareness of this subject could raise interest. 
To improve your manuscript you should focus in introduction more on the current knowledge of HG and any other proteins and what treatment is effective based on EBM. 

Response: We feel like we have addressed this issue. If the reviewer has specific areas of improvement, we are happy to incorporate them. We feel like this this has been done throughout the manuscript. As to not be redundant this has not been changed.

Comment 2: Regarding paragraph 3 - drug development - a table could be more useful to synthesize already known information regarding treatment options.

Response: We have added a table at the end of the 3. Drug Development for HG section.

Comment 3: In conclusion section you could also address study directions relating to screening women at risk for HG.

Response: A concluding sentence has been added to this point.